# Negative air ions through the action of antioxidation, anti-inflammation, anti-apoptosis and angiogenesis ameliorate lipopolysaccharide induced acute lung injury and promote diabetic wound healing in rat

**Yu-Hsuan Cheng[1]**, **Hung-Keng Li[1,2]**, **Chien-An Yao[3]**, **Jing-Ying Huang[1]**, **Yi-Ting Sung[1]**, **Shiu-Dong Chung[2,4,5]** *, **Chiang-Ting Chien**[1] *

1 Department of Life Science, School of Life Science, College of Science, National Taiwan Normal University, Taipei, Taiwan, 2 Division of Urology, Department of Surgery, Far-Eastern Memorial Hospital, New Taipei City, Taiwan, 3 Department of Family Medicine, National Taiwan University Hospital, Taipei, Taiwan, 4 Department of Nursing, College of Healthcare & Management, Asia Eastern University of Science and Technology, New Taipei City, Taiwan, 5 General Education Center, Asia Eastern University of Science and Technology, New Taipei City, Taiwan

☯ These authors contributed equally to this work.
* ctchien@ntnu.edu.tw (CTC); chungshiudong@gmail.com (SDC)

## Abstract

Negative air ions (NAIs) being bioactive and negative charged molecules may confer antioxidant and anti-inflammatory activity. We assessed the effect of NAIs on two inflammatory diseases in animal models including lipopolysaccharide (LPS) induced acute lung injury (ALI) and wound healing in diabetic rats. We used intra-tracheal infusion of LPS to induce ALI and made a full-thickness cutaneous wound in streptozotocin-induced diabetic female Wistar rats. We evaluated NAIs effects on reactive oxygen species amount, leukocyte infiltration, wound healing rate, western blot, and immunohistochemistry in the lungs of ALI and skin sections of wounds. Our data found NAIs exposed saline displayed higher antioxidant activity vs. non-exposed saline. NAIs exposure did not significantly affect arterial blood pressure and respiratory frequency in control and LPS treated groups. LPS increased leukocyte infiltration, caspase 3/Poly-ADP-ribose-polymerase-mediated apoptosis formation and decreased Beclin-1/LC3-II-mediated autophagy in lungs. NAIs exposure conferred pulmonary protection by depressed leukocyte infiltration and caspase 3/Poly-ADP-ribose-polymerase mediated apoptosis and enhanced LC3-II-mediated autophagy in LPS induced ALI. NAIs treatment resulted in a significantly accelerated wound closure rate, decreased erythrocyte accumulation and leukocyte infiltration mediated oxidative stress and inflammation, and upregulated expression of skin collagen, vascular endothelial growth factor receptor-2 (VEGFR-2) and factor transforming growth factor-beta 1 (TGF-β1) vs non-treated group. Based on these results, it is suggested that NAIs conferred a protection through the upregulating LC3-II-dependent autophagy mechanism and downregulating leukocyte infiltration mediated inflammation and caspase 3/Poly-ADP-ribose-polymerase signaling in the LPS-

**Data Availability Statement:** All relevant data are within the paper and its Supporting Information files.

**Funding:** This source of funding was partly supported by Ministry of Science and Technology of ROC with the funding number: MOST-107-2218-E-003-001 (Chiang-Ting Chien). The funders had no role in study design, data collection and analysis, decision to publish, or preparation of the manuscript.

**Competing interests:** The authors have declared that no competing interests exist.

treated ALI and promoted diabetic wound healing through the enhancing skin collagen synthesis, VEGFR-2 and TGF-β1 pathways.

## 1. Introduction

Air ions are electrically charged molecules or atoms present in the atmosphere. Negative air ions (NAIs) are negatively charged molecules that obtain electrons in the air [1, 10], whereas positive air ions (PAIs) lose electrons. NAIs are bioactive molecules and have been reported to affect the cardiovascular and respiratory systems. There were several studies to demonstrate the beneficial or therapeutic effects of NAIs exposure on lung function, metabolic measures, and asthmatic symptoms [2, 3]. The biological effects of air ions on indoor air quality as well as the various health advantages have been reported by Grinshpun et al. [4]. By use of mass spectrometers analysis, the major components of NAIs are $CO_3^-$, $O^-$, $O_3^-$, and $NO_3^-$ [5, 6]. Several methods can be applied to produce NAIs including radiation lines or cosmic rays in the air, ultraviolet rays in sunlight, natural and anthropogenic corona discharge, shearing forces of water/Lenard effect, and plant-based sources of energy [7]. Among indoor air purifiers of different types, ionic emitters have gained increasing attention and are presently used for removing dust particles, aeroallergens and airborne microorganisms from indoor air [4]. In the present study, we established the NAIs generator using artificial corona discharge in combination with electronic electrostatic precipitate collection.

NAIs have been found to inhibit growth of bacteria and fungi on solid media, to exert a lethal effect on vegetative forms of bacteria suspended in water, and to decrease the viable count of bacterial aerosols [8]. Particulate matter (PM) is an air pollutant that endangers human health [9]. Related researches have demonstrated that NAIs were effective in decreasing PM2.5 amount [10]. In biomedical application, Krueger et al. [11] indicated NAIs accelerate tracheal activity in mammals *in vitro* and *in vivo* and found that the hemoglobin of animals killed by carbon dioxide recovered its bright red color more quickly in NAIs than in ordinary air. Sirota et al. [12] showed that inhaling NAIs for 60 min at the dosage of 320–350 000 ions/$cm^2$ in animals activated the respiratory organs by acting directly on mucosal and blood, but their mechanism is unknown. Recently, the outbreak of coronavirus disease 2019 (COVID-19) caused by severe acute respiratory coronavirus 2 (SARS-CoV-2) has spread widely in the world [13]. SARS-CoV and SARS-CoV-2 could induce acute lung injury (ALI) and inflammation, which is mimicked by lipopolysaccharide (LPS), a bacterial endotoxin present on the outer membrane of Gram-negative bacteria [14], implicating SARS-CoV-2 and LPS through similar pathways of binding toll-like receptor-4 (TLR4) to activate angiotensin converting enzyme 2 (ACE2), to induce ALI and hyperinflammation [15, 16]. There has been an increase in the understanding of the mechanisms for induction of ALI, however, there have been no alternative treatments to reduce mortality rates in ALI patients [17]. Autophagy is a biological process that occurs in eukaryotes and maintains cell homeostasis and viability through phosphatidylinositol-3-kinase (PI3K)/AKT/the mammalian target of rapamycin (*mTOR*) signaling to activate protective Beclin-1/Atg5-Atg12/LC3-II signaling [16, 18]. Given the urgency for the protection of COVID-19 induced ALI, NAIs may be adapted to ameliorate LPS induced ALI.

On the other hand, chronic refractory wounds are a multifactorial comorbidity of diabetes mellitus (DM) with the characteristic of impaired vascular networks and increased oxidative stress and inflammation. Rat skin has been approached to evaluate the possible effect of PAIs or NAIs on wound healing. The data informed PAIs interfered with wound healing, while

NAIs accelerated wound healing [3]. However, the mechanism was not clarified. Kim et al. [19] demonstrated that NAIs treatment inhibited intracellular reactive oxygen species production, p38 mitochondrial protein initiator kinase activation, and activation protein 1 (c-Fos and c-Jun) activation in human keratinocyte cell line HaCaT implicating antioxidant and anti-inflammatory effects. Based on these information, this study was designed to explore the NAIs effect on two inflammatory animal models: 1) LPS induced ALI and inflammation in rats, and 2) wound healing of skins in DM rats.

## 2. Methods and materials

### 2.1 NAIs generation setup

NAIs generation was produced and supplied by the Orito Ecological Negative Ion Machine B (A-168 OLiDE Oriday Technology Co., Ltd., Banqiao District, New Taipei City, Taiwan). The principle for producing NAIs was demonstrated in **Fig 1A**.

### 2.2 Animal model of ALI

Female Wistar rats (200–250 g) were purchased from BioLASCO Taiwan Co. Ltd. (I-Lan, Taiwan) and housed at the Experimental Animal Center, National Taiwan Normal University, at a constant temperature and with a consistent light cycle (light from 07:00 to 18:00 o'clock). These animals were fed a standard chow (Laboratory Rodent Diet 5001 containing 0.4% sodium) and tap water ad libitum. All surgical and experimental procedures were approved by Institutional Animal Care and Use Committee of National Taiwan Normal University (Approval number 110012 on the date of 2021 August 5) and were in accordance with the guidelines of the National Science Council of Republic of China. All animal experiments were performed under anesthesia, all efforts were made to minimize suffering and finally sacrificed with overdose of anesthesia. These messages were described in detail in following experiments.

The animals were grouped as follows: n = 9 animals per group: Control (Con) group, LPS treated (LPS) group, Control group + NAIs treated (Con + NI) group and LPS treatment + NAIs treated (LPS + NI) group. Rats were anesthetized with subcutaneous urethane (1.2 g/ kg, Sigma-Aldrich, St. Louis, MO). Femoral venous catheter (PE-50; Clay Adams, Parsippany, NJ) was inserted to facilitate blood sampling and medication management. Vascular patency was maintained by intravenous infusion pump (Infors, Bottmingen, Switzerland) at a rate of 1.2 ml/min. A femoral artery was catheterized to measure arterial blood pressure by a polygraph (ML845 Power Lab 4/25 T System, ADInstruments, Sydney, Australia). During the preparation process, all rats recorded breathing frequency and arterial blood pressure with a polygraph. During the stabilization period of 30 min after anesthesia, 200 μg/kg of LPS stimulation was immediately infused into trachea at 200 μl for 210 min. Physiological functions were continuously recorded for 30 min of baseline period and 210 min of experimental period, and arterial blood gases and respiratory mechanics were recorded at the end of the experiment. The NAIs treatment to the rats by 10-cm distance of NAIs exposure were performed for 210 min. Control rats were also placed indoors, but are only exposed to the air in the environmental chamber. We determined the respiratory rate, flow rate and breathing depth in these animals in total 240 min. After experiments, the animals were sacrificed with overdose of urethane.

At different times after LPS stimulation (previously specified), the lungs were removed and immobilized, through the trachea at a pressure of 30 cm $H_2O$, in 24 h via the trachea in the form of 10% buffer. The lungs are then dehydrated by a series of ethanol solutions, cleared with methyl salicylate, and embedded in paraffin. Parts of the lung tissue (5 microns) were mounted on a glass slide and stained with hematoxylin and eosin (H&E) for light microscopy.

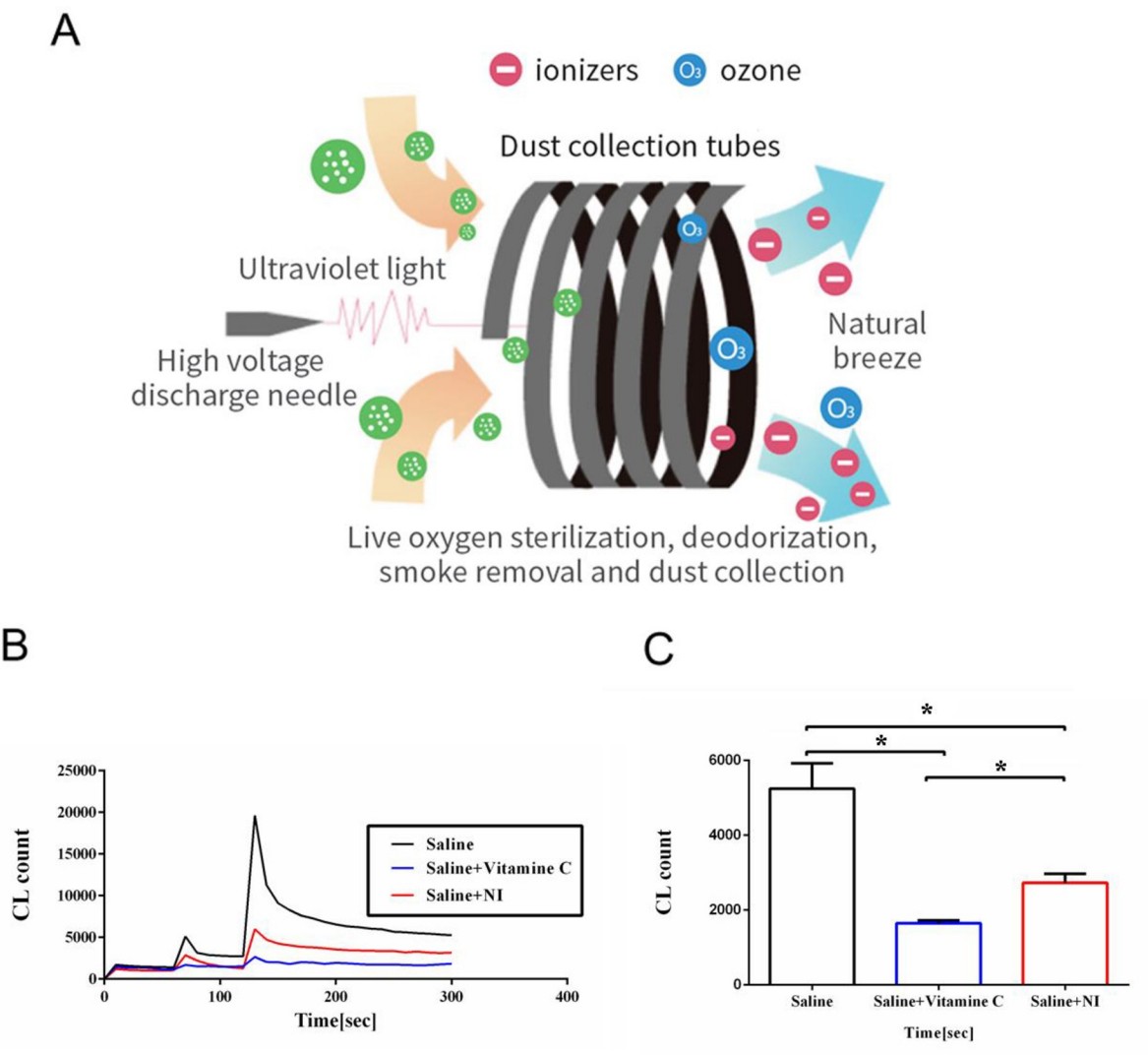

**Fig 1.** A: the principle of NAIs production in this study. B: three original data from an ultrasensitive chemiluminescence analyzer. C: statistical difference between saline and NAIs treated saline (Saline + Ni). Data were expressed as mean ± SEM (n = 6). * $P < 0.05$ between two groups comparision.

For immunochemical analysis, these sections were first treated with a blocking buffer for 15 min to block non-specific background binding. After phosphate buffer saline (PBS) flushing, the section is treated with A Zymed element (Zymed Laboratory in South San Francisco, CA). This included secondary biotin goat anti-rabbit serum, followed by marsh peroxide-tagged streptococcal and subsequently developed diamine amphetamine substrate solutions that produce a permanent color.

## 2.3 Hydrogen peroxide ($H_2O_2$) scavenging capacity of NAIs

In this part of study, the $H_2O_2$ scavenging activity of the saline with or without NAIs treatment was measured by a luminol-amplification ultrasensitive chemiluminescence detection technique as described in our laboratory [20]. In brief, 200 μL of saline with or without NAIs treatment was mixed with 0.5 mL of 0.1 mmol/L luminol (5-amino-2,3-dihydro-1,4-phthalazinedione, Sigma, Chemical Co., St. Louis, MO, USA) and 0.1 mL of $H_2O_2$ (0.03%)

and was analyzed with a chemiluminescence analyzing system (CLA-ID3, Tohoku Electronic Inc. Co., Sendai, Japan). The chemiluminescence signals emitted from the mixture of saline, $H_2O_2$ and luminol, which represented the hydrogen peroxide content in the mixture, were recorded. The increased chemiluminescent signals from the sample–luminol–$H_2O_2$ mixture were determined for 300 s. The total chemiluminescent counts were calculated by the area under the curve and were displayed as counts/10 s.

## 2.4 Bronchoalveolar lavage for neutrophils and reactive oxygen species (ROS) amount

We obtained bronchoalveolar lavage (BAL) of rats to detect oxidative stress and leukocytes as described previously in our laboratory [21]. In brief, after LPS stimulation, the chest cavity was opened, the left main-stem bronchus was ligated, and the right lung was lavaged with 1 aliquot of $Ca^{2+}$/$Mg^{2+}$-free phosphate buffer saline (pH 7.4). The aliquot volume was estimated to 35 ml/kg of body weight (approximately 90% of total lung capacity). The aliquot was instilled into the lungs 3 times before final collection. The BAL were also performed with a chemiluminescence analyzer to quantify ROS amount as described above. A cell counting analyzer (Sysmex TM K-1000; TOA Medical Electronics Co. Ltd., Kobe, Japan) was applied to measure the number of neutrophils present in the BAL samples.

## 2.5 Western blot of lung tissue

In this study, we used immunoblotting techniques to measure autophagy-related Beclin-1 and LC3-II, apoptosis-related caspase 3 and poly-(ADP-ribose)-polymerase (PARP) signaling pathway in the lung tissue. The methods have been well established and described previously in our laboratory [22]. In brief, twenty μg of proteins were used in 10% polyacrylamide gels with electronphoretically transferred to nitrocellulose membranes (Amersham Biosciences, Buckingham, England, UK). The membrane was blocked and incubated at 4˚C. The primary antibodies were applied to identify Beclin-1 (Cell Signaling Technology, Inc., Danvers, MA, USA), LC3-II (Cell Signaling Technology), the activation fragments (17 kDa of cleaved product) of caspase 3 (CPP32/Yama/Apopain, Upstate Biotechnology, Lake Placid, NY), PARP (Cell Signaling Technology), and β-actin (Sigma). The density of protein bands and the appropriate molecular mass were semi-quantitatively analyzed using an image analysis system (Alpha Innotech, San Leandro, CA, USA).

## 2.6 In situ detection of apoptosis and autophagy formation in the lung

We assessed the degree of apoptosis by evaluating the expression of caspase 3 and PARP and terminal deoxynucleotidyl transferase-mediated nick-end labeling (TUNEL) assay (DNA Fragmentation Kit; BioVision, Milpitas, California, USA) and autophagy by Beclin-1 (Cell Signaling Technology, Inc.) and LC3-II (Cell Signaling Technology, Inc.) staining in the rat lungs. The determined techniques have been followed as reported previously [22]. In brief, tissue sections from 10% formalin fixation and paraffin embedding were deparaffinized, rehydrated, and stained with H&E or immunohistochemically. For autophagy or apoptosis related proteins staining, the 5-μm tissue sections (Leica RM 2145, Nussloch, Germany) were incubated overnight at 4˚C with mouse anti-rat Beclin-1 antibody (BD Biosciences, San Jose, CA, 1:100) or LC3-II (Cell Signaling Technology, Inc., 1:100), Caspase 3 (CPP32/Yama/Apopain, Upstate Biotechnology, Lake Placid, NY, 1:100), PARP (Cell Signaling Technology, Inc., 1:100). Subsequently, biotinylated secondary antibodies (Dako, Botany, NSW, Australia) were applied, followed by incubation with streptavidin-conjugated horseradish-peroxidase (Dako). The chromogen used in this study was Dako Liquid diaminobenzidine. Twenty high-power (×400)

fields were randomly selected from each gastric section, and the value of brown deposits/total section area for Beclin-1, LC3-II, caspase-3 or PARP stain was analyzed with Adobe Photoshop 7.0.1 image software. TUNEL was performed according to a previously described method [16]. Briefly, 5-μm-thick sections of gastric tissues were prepared, deparaffinized, and stained using the TUNEL assay kit. Twenty high-power (×400) fields were randomly selected for each section, and the level of each oxidative stress was analysed with a Sonix Image Setup (Sonix Technology Co., Ltd., Hinschu, Taiwan, ROC).

## 2.7 Type I diabetes induction

The protocol for induction of diabetes was demonstrated previously [23]. We used intraperitoneal streptozotocin (STZ at 65 mg/kg; Sigma, Missouri, USA) to induce type I DM. We determined the body weight and fasted blood glucose before and after STZ treatment every week. After one week of STZ injection, the fasted (8 h) blood glucose level was examined by using Contour plus blood glucose monitoring system (Bayer, Leverkusen, Germany). The fasted blood glucose > 250 mg/dL was identified as successful diabetes induction. Under 2% isoflurane anesthesia, the skin hair on the rat back was shaved off and cleaned with iodine wine, and the post-disinfection surgical scissors created a square full cortex wound of 3 cm × 3 cm in size (cortex thickness< 3 mm) in the center of the rat's back. The rats were given standard feed and povidone iodine to prevent wound infection. For analgesia, 20 mg/kg dose paracetamol was added to daily drinking waters of the rats in all groups. Wound closure was measured at 0, 3, 7, 10, 14, and 21 days after injury. The animals were euthanized with overdose of urethane on days 7, 14 and 21 and wound samples from adjacent skins were collected, fixed in 10% paraformaldehyde for histological examination or quick-frozen in liquid nitrogen and stored at -80°C for further analysis.

## 2.8 Grouping for wound healing

Experimental rats were divided into: (1) Control (Con) group: normal rats that only created full cortex wounds on the back and did not receive any treatment; (2) Diabetes mellitus group (DM group): diabetic rats induced by STZ were made a full-skin wound on the back; (3) Control rats with NAIs Group (Con + NI group): healthy rats treated with NAIs and made a full cortex wound on the back and (4) Diabetes mellitus with NAIs treatment group (DM + NI group): diabetic rats treated with NAIs after manufacturing a full cortex wound on the back. Total 24 rats were used and each group was 6 animals.

Rats in the NAIs treated group were placed under the Oridge Ecological Negative Ion Machine B on a standard 45.5 × 23 × 20.5 cm$^3$ experimental cage, and blowed the wound with NAIs for one hour every day. The rats in the sham surgery group were kept in another chamber to avoid interference from NAIs. On the 0th, 3rd, 7th, 11th, 14th and 21st days after wound manufacturing, the wound healing situation was recorded.

## 2.9 Histopathology of rat skins

Rat skins were prepared with H&E for morphologic analysis, and with Masson's Trichrome and Sirius red stains to visualize collagen organization. The protocols of histopathology were described previously in our laboratory [16]. In brief, the tissues were collected, treated with formaldehyde (10%), and embedded in paraffin wax. The samples were then cut into sections with a thickness of 5 μm (Leica RM 2145, Nussloch, Germany). After the tissue sections had been waxed and hydrated in the usual way, the sections were stained with Masson's Trichrome staining kit (HT15, Sigma, USA). With this staining, the collagen was dyed blue and the keratin was dyed in red color. Other slides were subjected to Sirius Red 80 (Sigma Aldrich; 0.1% of

Sirius Red in saturated aqueous picric acid) staining. The stained samples were then examined using a light microscope.

## 2.10 Immunohistochemistry (IHC) for TGF-β1 and VEGFR-2

Skin is capable of spontaneous self-repair following injury. These functions are mediated by numerous pleiotrophic growth factors, including the vascular endothelial growth factor-2 (VEGFR-2) and transforming growth factor-β1 (TGF-β1) families [24, 33]. We assessed the effect of NAIs on the expression of the VEGFR-2 and TGF-β1 in skin wound healing of rats. The ligands of VEGFR-2 were secreted by keratinocytes and macrophages and increased VEGFR-2 performance, increased angiogenesis, provided more nutrients and oxygen to the wound, and promoted wound healing [24]. The latter two were secreted in large quantities by macrophages, which have the effects of angiogenesis and wound healing via collagen reconstruction. In addition, according to previous studies, VEGFR-2 can increase the wound healing area, and slow down the infiltration of immune cells between tissues and the formation of scars [25]. In the present study, the skin sections of different groups were determined by performing TGF-β1 and VEGFR-2 IHC. In brief, tissue sections from 10% formalin fixation and paraffin embedding were deparaffinized, rehydrated, and stained with VEGFR-2 or TGF-β1. For VEGFR-2 or TGF-β1 staining, the 5-μm tissue sections were incubated overnight at 4˚C with mouse anti-rat VEGFR-2 antibody (VEGF #9698 Cell Signaling Technology, Danvers, MA, USA, 1:100) or TGF-β1 (sc-146 Santa Cruz, Dallas, Texas, USA, 1:100). Next, biotinylated secondary antibodies (Dako, Botany, NSW, Australia) were applied, followed by incubation with streptavidin-conjugated horseradish-peroxidase (Dako). The chromogen used in this study was Dako Liquid diaminobenzidine. Twenty high-power (×400) fields were randomly selected from each section, and the value of brown deposits/total section area for VEGFR-2 or TGF-β1 stain was analyzed with Adobe Photoshop 7.0.1 image software as described above.

## 2.11 Wound preparation

Under averin anesthesia, the skin hairs on the back was shaved off and cleaned with iodine wine. By using the surgical scissors, the rat's back in the center was created a square full cortex wound of 3 cm × 3 cm in size (cortex thickness < 3 mm). The degree of wound closure was determined at 0, 3, 7, 11, 14, and 21 days after injury. Animals were euthanized on days 7, 14 and 21 and wound samples and adjacent normal skin were collected, fixed in 10% paraformaldehyde for histological examination or quick-frozen in liquid nitrogen and stored at -80˚C for further analysis. Wound healing rate was calculated the percentage of wound closure by the initial and final areas plotted on the slide during the experiment. Wound shrinkage is calculated as follows [26]:

A0 was the original wound area and At was the wound area at the time of biopsy on days 0, 3, 7, 11, 14 and 21, respectively.

## 2.12 Statistical analysis

All values are expressed as means ± SEM unless stated otherwise. Between-group comparisons were performed by using unpaired $t$ tests or analysis of variance with Bonferroni method as post hoc analysis; within-group comparisons were performed by using paired t-tests or repeated-measures analysis of variance with Bonferroni method as post hoc analysis. A value of $P < 0.05$ indicated statistical significance. All computations were performed with SPSS for WINDOWS software (version 13.0; SPSS Inc, Chicago, IL).

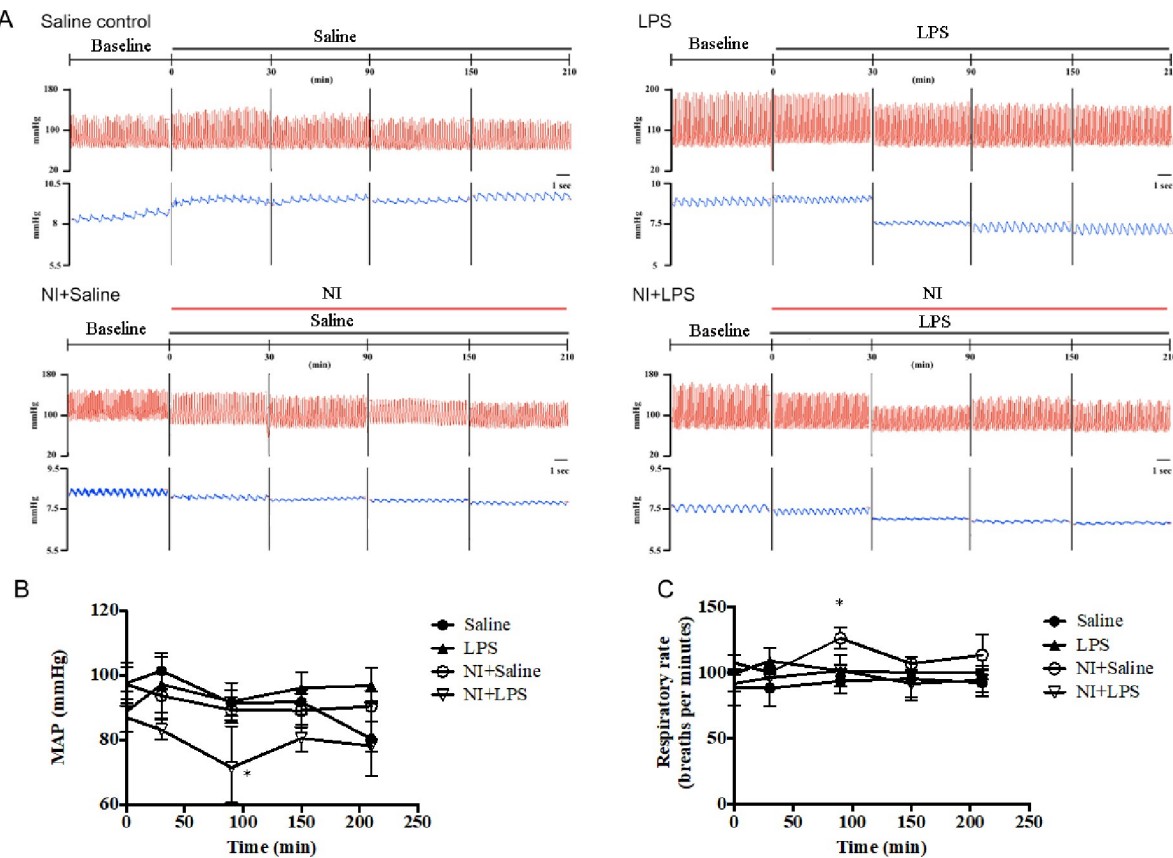

**Fig 2. The physiologic parameters including mean blood pressure (MAP) and respiratory rate in response to LPS or NAIs in four groups of rats.** A: the original data from four groups of rats. B: the statistical data of MAP in four groups of rats. C: the statistical data of respiratory rate in four groups of rats. Data were expressed as mean ± SEM (n = 9). * $P < 0.05$ vs. Saline groups.

## 3. Results

The principle of NAIs generation was described in **Fig 1A**. In vitro experiment to determine NAIs effect on the antioxidant activity, the saline was exposed to NAIs for one hour. Our results demonstrated the original data from the chemiluminescent analyzer in **Fig 1B** and found a markedly depressed ROS response in the NAIs treated saline as compared to non-treated saline. The statistical data indicated that NAIs exposed saline displayed a significant decrease in ROS level vs. the non-treated saline (**Fig 1C**).

We first explored the NAIs effect on mean arterial blood pressure and respiratory frequency in response to LPS-induced ALI for 210 min. As shown in **Fig 2A**, the representative graphs of physiological parameters were displayed in four groups of rats. The statistical data of the changes of mean arterial blood pressure (**Fig 2B**) and respiratory frequency (**Fig 2C**) in four groups of rats treated with LPS with or without NAIs exposure did not show any very significant differences of cardiorespiratory parameters in 210 min of LPS stimulation. These data implicate that NAIs have no effect on cardiorespiratory parameters during our acute challenge.

We determined the effect of NAIs on autophagy and apoptosis in LPS-induced ALI by evaluating the protein levels of Beclin-1/LC3-II, and caspase 3/PARP signaling with Western blotting and IHC. As shown in **Fig 3A** of the original Western blot, the expression of caspase 3 was significantly enhanced in the LPS group vs. Con group, whereas cleavage caspase 3 was

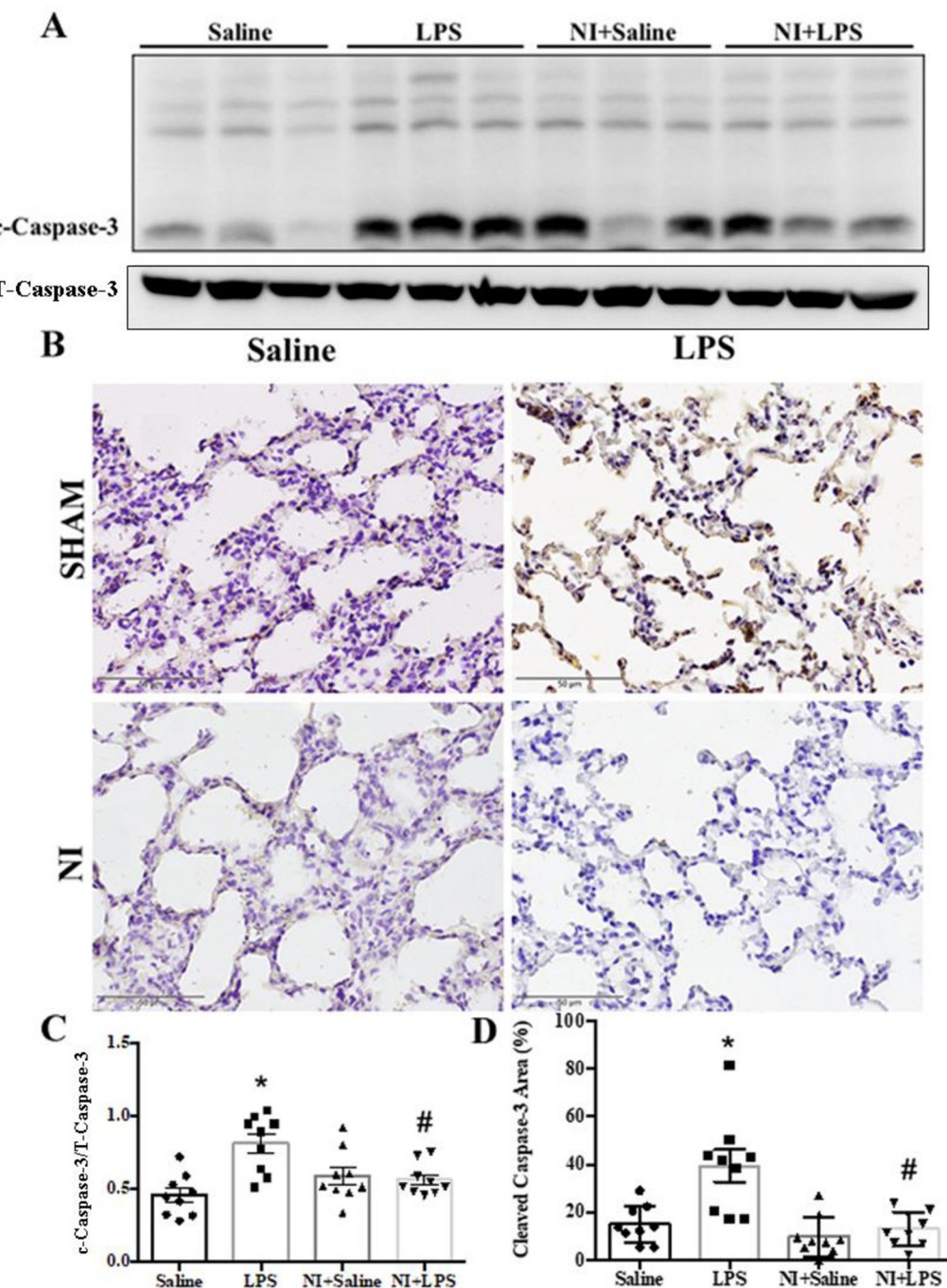

**Fig 3. Effect of 60-min NAIs treatment on c-caspase 3 expression in LPS induced ALI for 210 min of four groups of rats.** A: original western blot of caspase 3. B: typical caspase 3 immunohistochemic graphs of four groups of rat lungs. C: Statistical data of caspase 3 western blot. D: Statistical data of pulmonary immunohistochemic caspase 3 stains. Data are expressed as mean ± SEM (n = 9). * $P < 0.05$ vs. Saline group. # $P < 0.05$ vs. LPS group.

decreased in the LPS + NI group vs. LPS group (**Fig 3C**). In IHC analysis, LPS stimulation could increase the level of caspase 3 expression in our LPS-induced ALI model (**Fig 3B**), whereas NAIs exposure significantly inhibited cleavage caspase 3 in LPS-induced ALI (**Fig 3D**). As shown in **Fig 4A** of the original Western blot, the expression of PARP was significantly

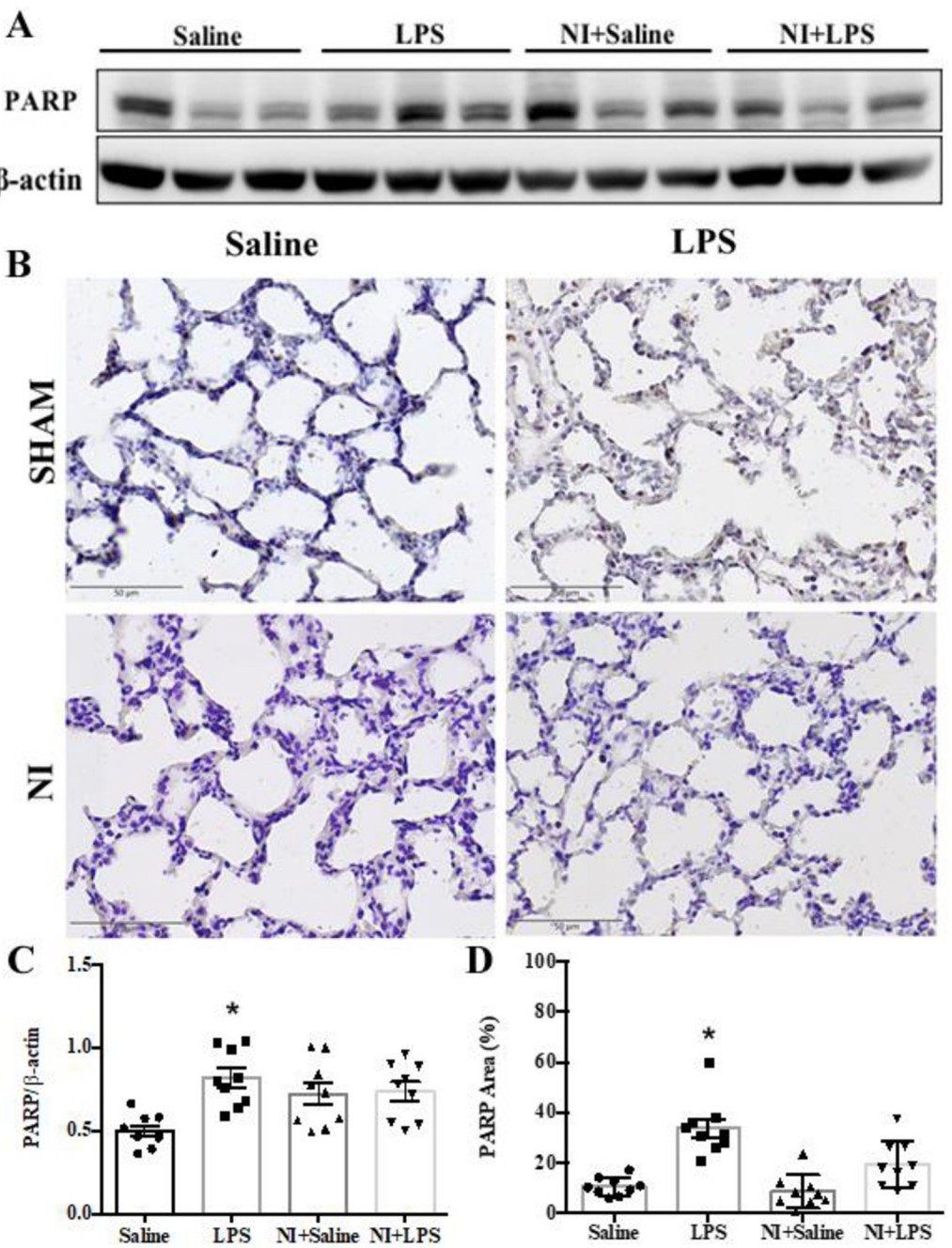

**Fig 4. Effect of 60-min NAIs treatment on PARP expression in LPS induced ALI for 210 min of four groups of rats.** A: original western blot of PARP. B: typical PARP immunohistochemic graphs of four groups of rat lungs. C: Statistical data of PARP western blot. D: Statistical data of pulmonary immunohistochemic PARP stains. Data are expressed as mean ± SEM (n = 9). $^*$ $P < 0.05$ vs. Saline group. # $P < 0.05$ vs. LPS group.

enhanced in the LPS group vs. Con group, whereas PARP expression was decreased in the LPS +NI group vs. LPS group (**Fig 4C**). In IHC analysis, LPS stimulation could also increase the level of PARP expression in our LPS-induced ALI model (**Fig 4B**), whereas NAIs exposure significantly inhibited PARP expression in LPS-induced ALI (**Fig 4D**). We further determined the leukocyte infiltration and TUNEL stain in the four groups of rat lungs with ALI. In **Fig 5A**,

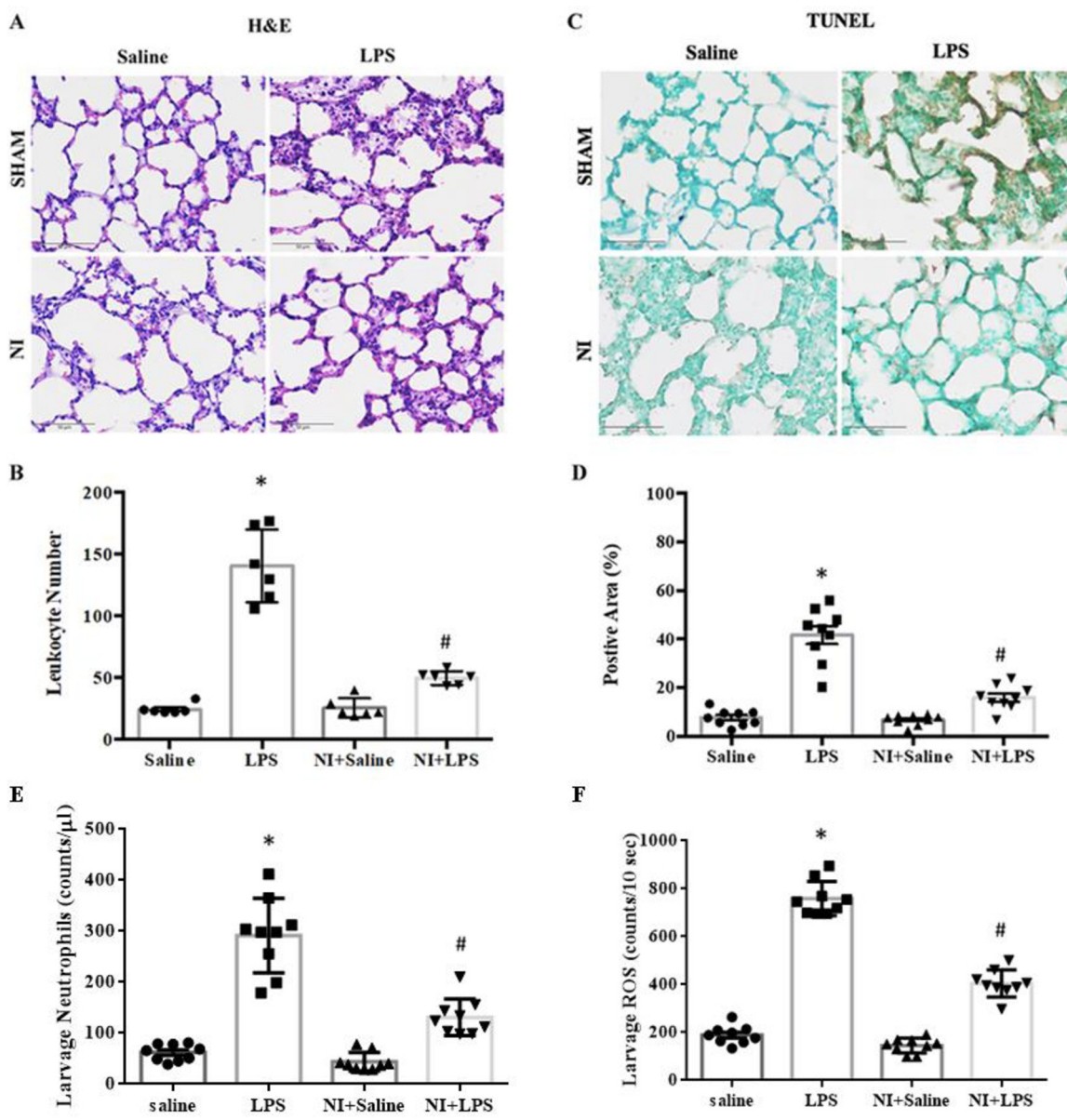

**Fig 5. Effect of 60-min NAIs treatment on leukocyte infiltration, TUNEL expression, larvage neutrophils and larvage ROS in LPS induced ALI for 210 min of four groups of rats.** A: H&E stained lung sections of four groups of rats. B: statistical data of leukocyte infiltration of four groups of rat lungs. C: original graphs of TUNEL stains of four groups of rat lungs. D: Statistical data of pulmonary TUNEL stains of four groups of rats. E: Larvage neutrophils number and F: Larvage ROS amount in four groups of rats. Data are expressed as mean ± SEM (n = 9). * $P < 0.05$ vs. Saline group. # $P < 0.05$ vs. LPS group.

H&E analysis showed that the marked increase of leukocytes infiltration was found in the LPS treated lung, whereas other three groups of lungs displayed a decrease of leukocyte infiltration. The statistical data demonstrated that the degree of leukocyte infiltration was significantly increased in the LPS group, whereas NAIs treatment significantly decreased LPS-enhanced leukocyte infiltration (**Fig 5B**). In **Fig 5C**, TUNEL stains showed that the marked increase of TUNEL positive stained cells was noted in the LPS treated lung, whereas other three groups of lungs displayed a decrease of TUNEL positive cells in the lung. The statistical data showed that the degree of TUNEL-apoptosis was significantly increased in the LPS group, whereas NAIs

treatment significantly decreased LPS-increased TUNEL apoptosis (**Fig 5D**). Larvage neutrophils number in LPS group was significantly increased as compared to saline control group (**Fig 5E**), whereas NAIs treatment significantly decreased LPS-increased larvage neutrophils number in LPS + NI group. In addition, larvage ROS amount in LPS group was also significantly increased as compared to saline control group (**Fig 5F**), whereas NAIs treatment significantly decreased LPS-increased larvage ROS amount in LPS + NI group.

As shown in **Fig 6A** of the original Western blot, the expression of Beclin-1 was significantly suppressed in the LPS group vs. Con group, whereas Beclin-1 expression was partly recovered in the LPS + NI group vs. LPS group (**Fig 6C**). In IHC analysis, LPS stimulation could also decrease the level of Becin-1 expression in our LPS-induced ALI model (**Fig 6B**), whereas NAIs exposure significantly increased Beclin-1 expression in LPS-induced ALI (**Fig 6D**). We further determined the LC3-II Western blot and IHC in four groups of rat lungs with ALI. As shown in **Fig 7A** of the original Western blot, the expression of LC3II was significantly suppressed in the LPS group vs. Con group, whereas LC3II expression was partially recovered in the LPS +NI group vs. LPS group (**Fig 7C**). In IHC analysis, LPS stimulation consistently decreased the level of LC3II expression in our LPS-induced ALI model (**Fig 7B**), whereas NAIs exposure significantly increased LC3II expression in LPS-induced ALI (**Fig 7D**). Our results informed that NAIs enhanced autophagy-related protective mechanisms in LPS-induced ALI mouse model.

The blood glucose levels of DM rats were above $> 250$ mg/dL three days after injection and remained stable throughout the experiment. A total of 2 rats died during the model-establishment process and the induction success rate was 95%. The body weight of rats was measured on 0, 3, 7, 11, 14 and 21 days, and the weight of all four groups fell between 300–450 g. Rat weight, which was measured in all rats at set time points, initially increased slowly and then showed negative growth three weeks after injection (unpresented data). All the rats showed obvious symptoms of polydipsia, accompanied by polyuria, polyphagia and weight loss.

Representative images of wound area in each group at 0, 3, 7, 11, 14 and 21 days after surgery were shown in **Fig 8A**. Full-thickness skin wounds on the backs of diabetic rats were created and treated with NAIs or non-NAIs air. On day 3 after wound injury, there was no significant differences in wound healing rate among the groups. However, the wound area was significantly smaller in the Con + NI and DM + NI groups than in the DM group on days 7, 11, 14 and 21 ($P < 0.05$) (**Fig 8B and 8C**). On the days 7 after the manufacture of the full cortex wound on the back, it can be seen that the wound area of the Con + NI group and the DM + NI group was smaller than that of the untreated Con and DM group (**Fig 8A**). The difference in wound healing between the treatment group and the untreated group was more obvious at days 11, the wound shape of the untreated Con and DM groups had obvious observable unevenness, and the wound in the Con group was still covered with thick scabs. The scab condition of the wound in the DM group was still unstable, and the tissue fluid exuded continuously, while the wound area of Con + NI group and the DM + NI group was significantly smaller than that in the DM group on Day 11, the wound edge was flat and the thick scab shedding was perfect and the wound remains dry continuously, with no exudation of tissue fluid. At the days 21, the wounds in the Con, DM and DM + NI groups still had unheated scabs (considered to be unhealed wounds), but the unhealed wound area in the Con and DM groups was larger than that in the DM + NI group in the treatment group, and the wounds in the Con + NI group in the other treatment group were completely healed (**Fig 8A**). If the unhealed wound area of each group within 0, 3, 7, 11, 14, 21 days is superimposed into a graph (**Fig 8B**) presented, red represents the extent of the initial wound manufacturing, yellow, green, light blue, dark blue, pink in order represents the remaining unhealed range of the wound at 3, 7, 11, 14 and 21 days, it can be more obvious that the degree of wound healing in different groups

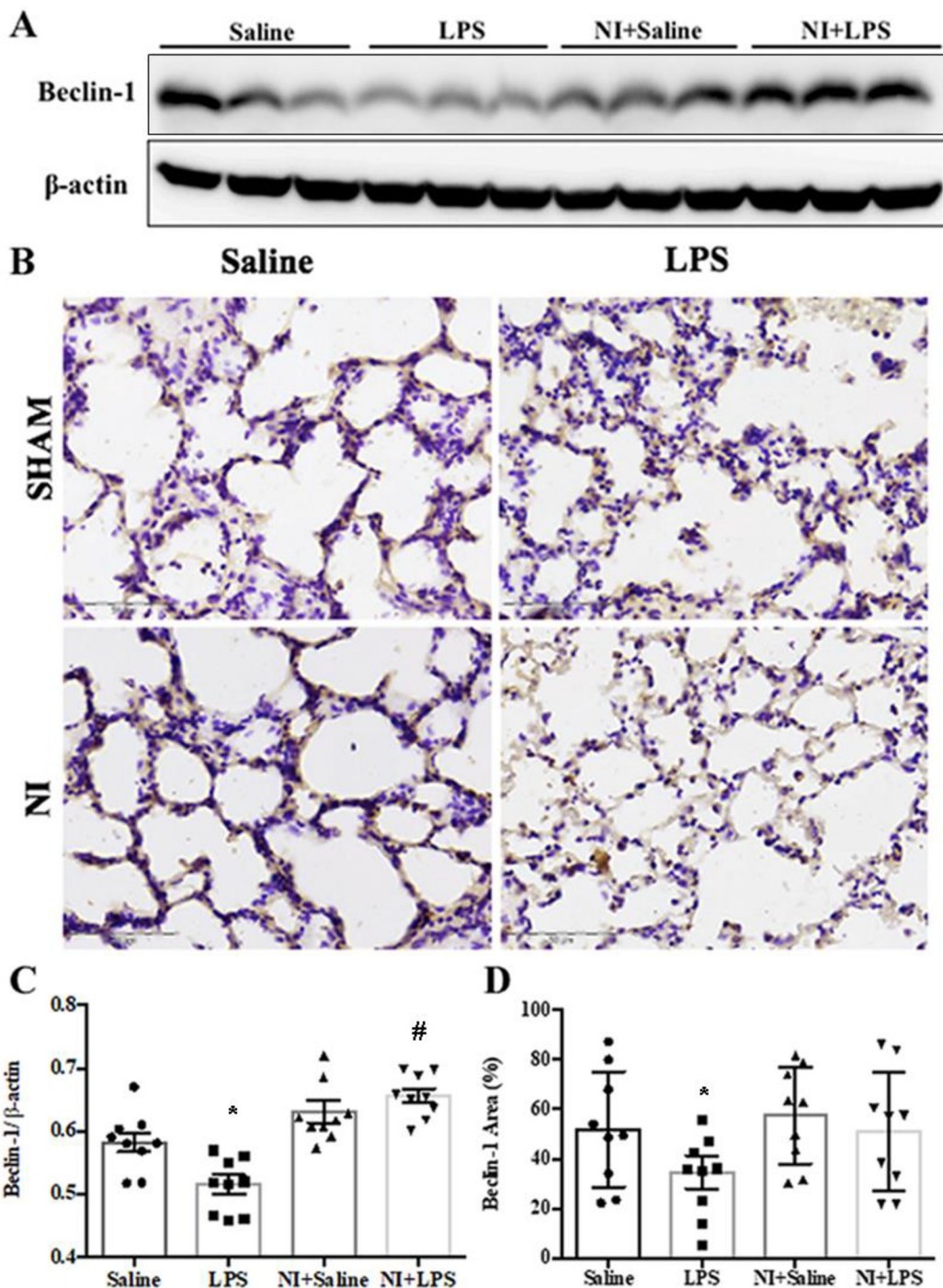

**Fig 6. Effect of 60-min NAIs treatment on Beclin-1 expression in LPS induced ALI for 210 min of four groups of rats.**
A: original western blot of Beclin-1. B: typical Beclin-1 immunohistochemic graphs of four groups of rat lungs. C: Statistical data of Beclin-1 western blot. D: Statistical data of pulmonary immunohistochemic Beclin-1 stains. Data are expressed as mean ± SEM (n = 9). * $P < 0.05$ vs. Saline group. # $P < 0.05$ vs. LPS group.

at different days. According to **Fig 8C**, the remaining unhealed wound area is small, and the NAIs treated group has the largest healing rate in the Con + NI group at days 21. Only < 1% of the wounds in the Con + NI group are still scabs (regarded as unhealed). The wound healing

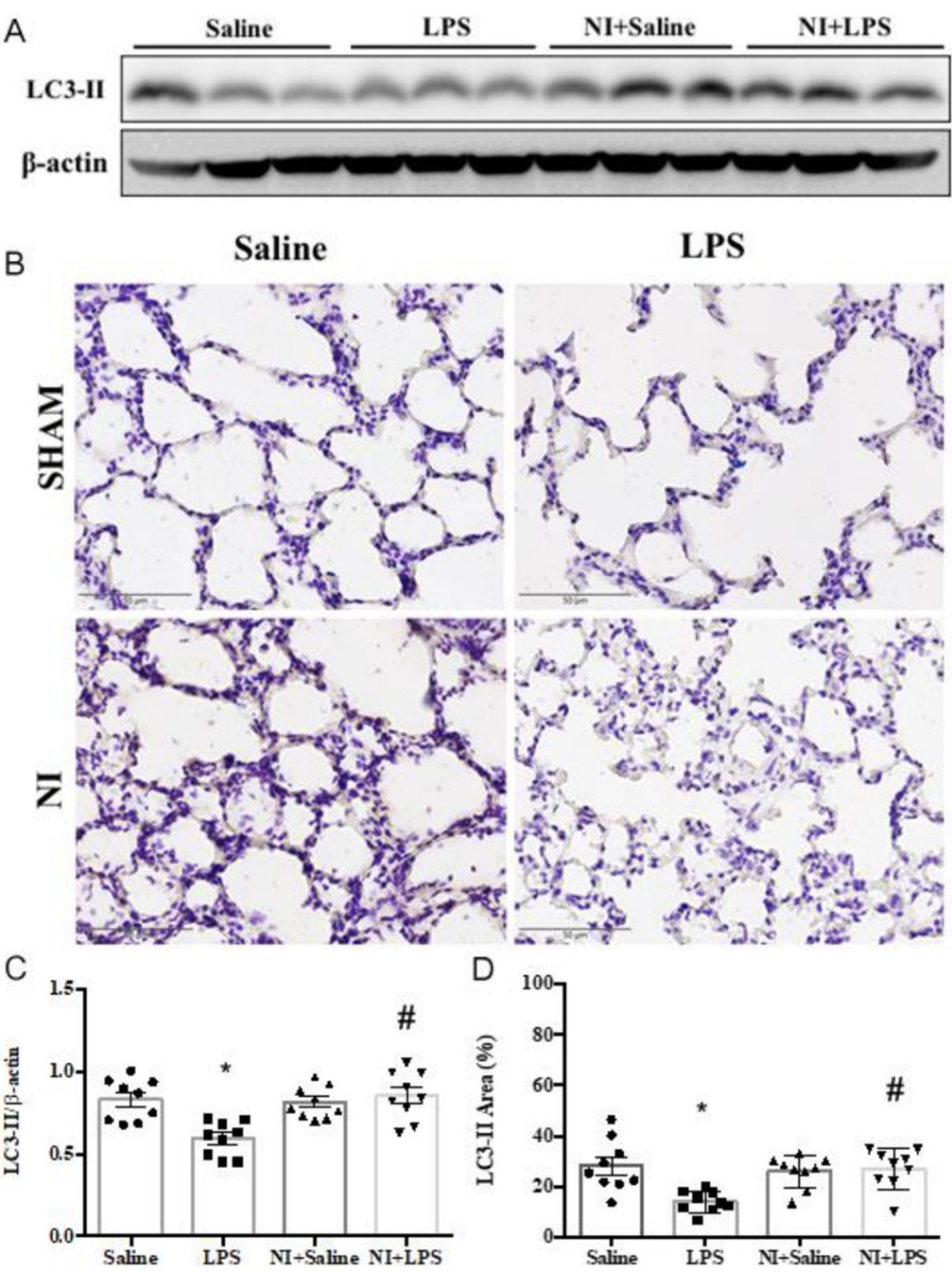

**Fig 7. Effect of NAIs treatment on LC3-II expression in LPS induced ALI of four groups of rats.** A: original western blot of LC3-II. B: typical immunohistochemic graphs of four groups of rat lungs. C: Statistical data of LC3-II western blot. D: Statistical data of pulmonary immunohistochemic LC3-II stains. Data are expressed as mean ± SEM (n = 9). * $P < 0.05$ vs. Saline group. # $P < 0.05$ vs. LPS group.

rate was higher in DM + NI group than DM group (**Fig 8C**). This finding suggests that the NAIs confer beneficial effects in increasing wound healing rate.

The histological structure of the regenerated dermis was analyzed on day 21. As shown in **Fig 9A** and **9D**, the epidermis of the new granulation tissue was intact and thick in all groups. In the Con + NI and DM + NI groups, new hair follicle formation was evident in the center of

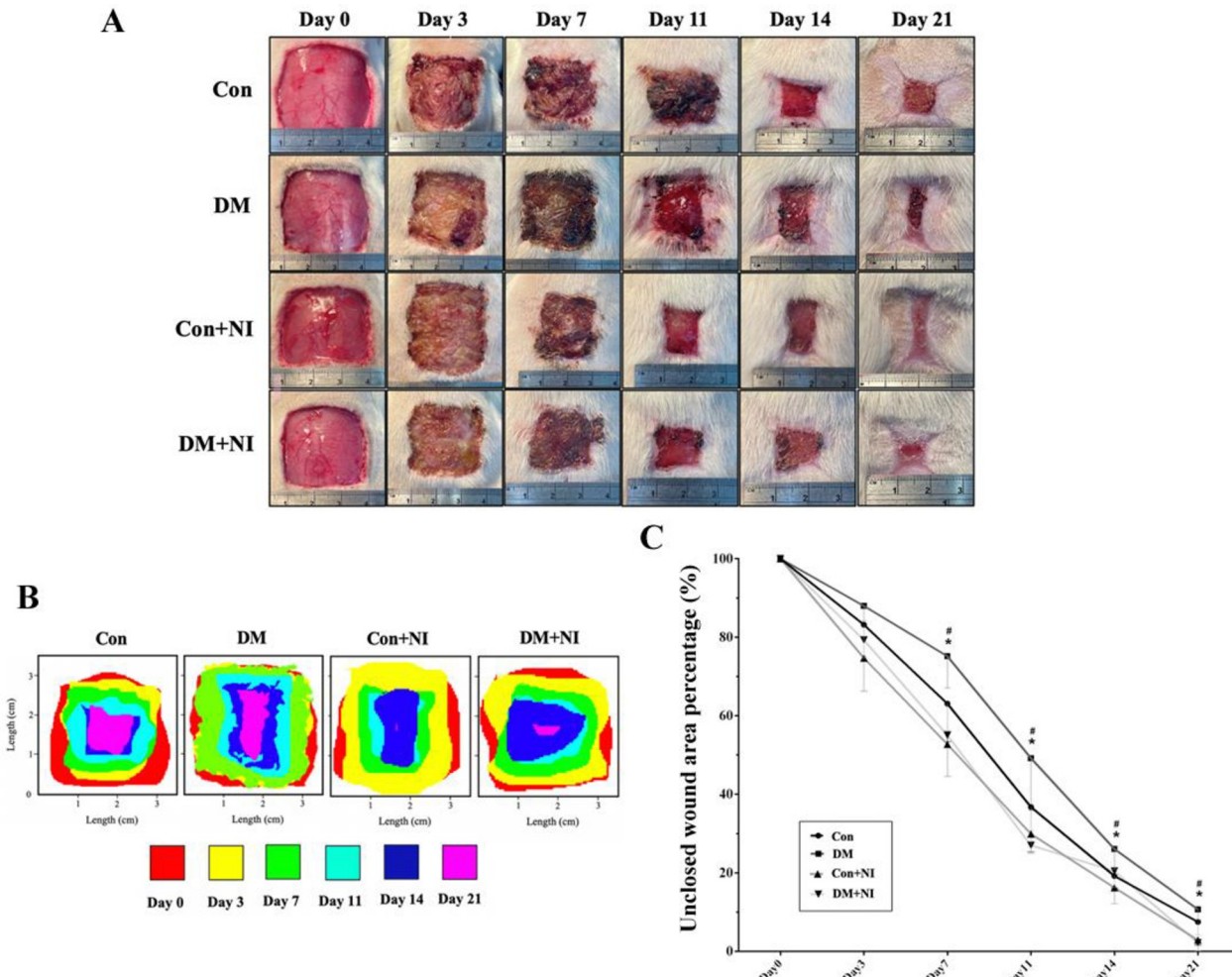

**Fig 8. Effect of NAIs exposure on the wound healing rate of rat skin of four groups of rats.** A: Representative images of the wound surface in each group on days 0, 3, 7, 11, 14 and 21. B: The merged images of different color correlated different time frame of wound area in four groups of rats. At day 21, the pink colored area is markedly reduced in the Con + NI group and DM + NI group. C: Statistical data of % unclosed wound area over time of four groups of rats. Data are expressed as mean ± SEM (n = 6). $^*$ $P < 0.05$ vs. Con group. # $P < 0.05$ vs. DM group.

the wound surface, and collagen deposition by Masson's stain was sufficient and orderly. In the DM group, the significant increase of RBC accumulation (**Fig 9A and 9B**) and inflammatory cell infiltration (**Fig 9A and 9C**) were observed at day 21. However, the degree of RBC accumulation (**Fig 9B**) and leukocyte infiltration (**Fig 9C**) was significantly depressed in the NI + NI and DM + NI groups.

The collagen deposition of four groups determined by Masson Trichrome's stain was shown in **Fig 9D**. A marked decrease in blue collagen deposition was found in DM group vs. other three groups. NAIs treatment in DM + NI group or Con + NI group displayed a significant blue collagen deposition compared to DM group (**Fig 9E**). The collagen deposition of four groups determined by Sirius red stain was also displayed in **Fig 10A**. A marked decrease in red collagen deposition was found in DM group vs. other three groups. NAIs treatment in DM + NI group or Con + NI group displayed a significant red collagen deposition as compared to DM group (**Fig 10B**).

The IHC results of the vascular TGF-β1 showed that the TGF-β1 exhibited in the Con + NI and DM + NI vascular interstitiums after NAIs treatment was higher than in the untreated

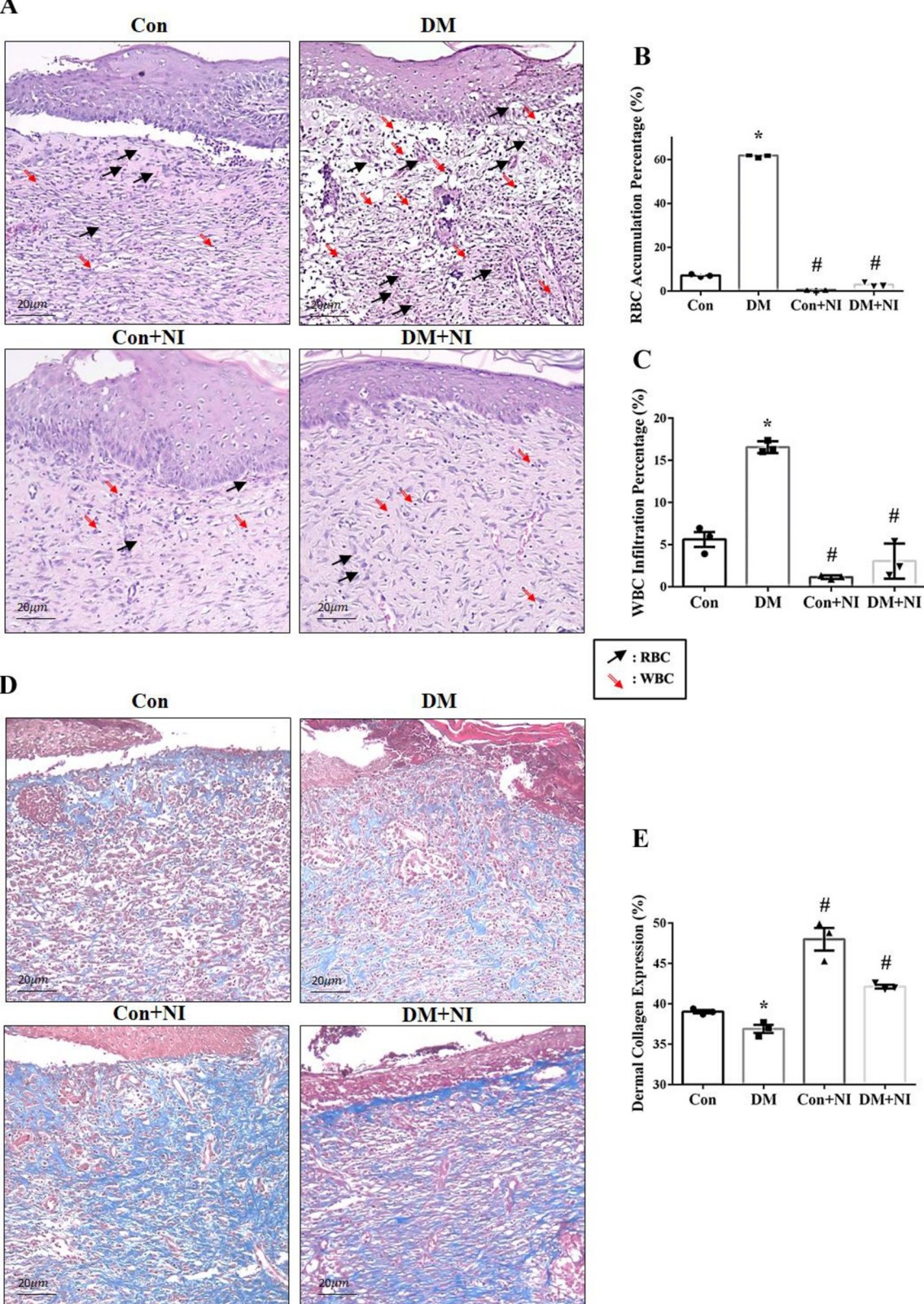

**Fig 9. Effect of NAIs exposure on H&E and Masson's trichrome stain in rat skin in the wound healing of four groups of rats.** A: Representative images of H&E histology on wound sections at day 21 (black arrow: WBC; red arrow: RBC). B: Analyses of RBC

accumulation of various treatment groups on day 21 (n = 3). C: Analyses of leukocyte infiltration of various treatment groups on day 21 (n = 3). D: Representative images of Masson's trichrome stain (blue color) on wound sections at day 21. E: Analyses of dermal collagen expression by Masson's stain of various treatment groups on day 21 (n = 3). Data are expressed as mean ± SEM (n = 3). * $P < 0.05$ vs. Con group. # $P < 0.05$ vs. DM group.

Con and DM groups (**Fig 11A and 11B**), and the TGF-β1 expression was statistically significantly less from the DM group compared with the Con group ($P < 0.05$). In contrast, there was no statistically significant difference in the vascular interstitial manifestations of TGF-β1 in the Con + NI and DM + NI groups after NAIs treatment indicating that the TGF-β1 expression after NAIs treatment could be restored to similar to that of the original Con group (**Fig 11A and 11B**). The IHC results of vascular VEGFR-2 expression showed that after NAIs treatment, the VEGFR-2 expression of the vascular beds in the Con + NI and DM + NI groups was higher than that in the untreated Con and DM groups (**Fig 11A and 11C**), and the vascular VEGFR-2 expression in the Con + NI and DM + NI groups had a statistically higher ($P < 0.05$) vs. DM group (**Fig 11C**).

## 4. Discussion

Ionic air purifiers have become widely and increasingly popular for removing dust particles, aeroallergens and airborne microorganisms from indoor air in various settings. In addition to the indoor air cleaning effect, the effect of NAIs has been determined by several investigators, however, there are still controversial results (favorable and unfavorable) about the performance of commercially available ionic air purifiers [4]. One paper showed that NAIs created by water shearing method improve aerobic metabolism only during a 1-h exposure, which may be caused by improvement of erythrocyte deformability, but NAIs created by corona discharge have no effects [27]. A potential mechanism is that NAIs enter the circulating blood via the lungs and electrons of these ions are delivered to the plasma protein [27]. The possible compositions of NAIs could be identified as $CO_3^-$ and $O^-$, $O_3^-$, $NO_3^-$, etc by using mass spectrometers [5, 6]. The antioxidant activity of erythrocyte cytosolic superoxide dismutase from rat, bovine, man and duck was increased when measured after NAIs exposure [28]. The primary physicochemical mechanism of biological action of NAIs is suggested to be related to the increase of superoxide dismutase activity by a low level of micromolar concentrations of $H_2O_2$ stimulation [28]. Our data using NAIs treated saline at a 10-cm distance with NAIs amount $> 1.6 \times 10^8/cm^3$ displayed a more efficient scavenging $H_2O_2$ activity vs. non-treated saline (**Fig 1C**) implicating the direct antioxidant activity by NAIs. These information conferred the hypothesis that the markedly antioxidant effect may have beneficial and biological effects on the inflammatory related diseases.

Until now, there was no strong evidence to clearly support a beneficial role of NAIs exposure on respiratory function or asthmatic symptom alleviation [29]. Liu et al. [30] observed no significant changes in cardiorespiratory biomarkers following the use of NAIs for 1 week. However, one previous research has reported that exercise-induced asthma was significantly attenuated by exposure to NAIs [2] implicating the NAIs' beneficial effect in respiratory system. Liu et al. [31] demonstrated the molecular linkages between indoor NAIs, PM reduction and cardiorespiratory function among children. They found that the increased NAIs associated with the decreased PM improved respiratory function mainly by eight pathways including promoting energy production, anti-inflammation and anti-oxidation capacity [31]. NAIs were also observed to be an energy-efficient air purification intervention that can effectively reduce the small airway particle exposure when a sufficient NAIs concentration is obtained [32]. ALI is a syndrome of pulmonary inflammation and permeability characterized by gas exchange

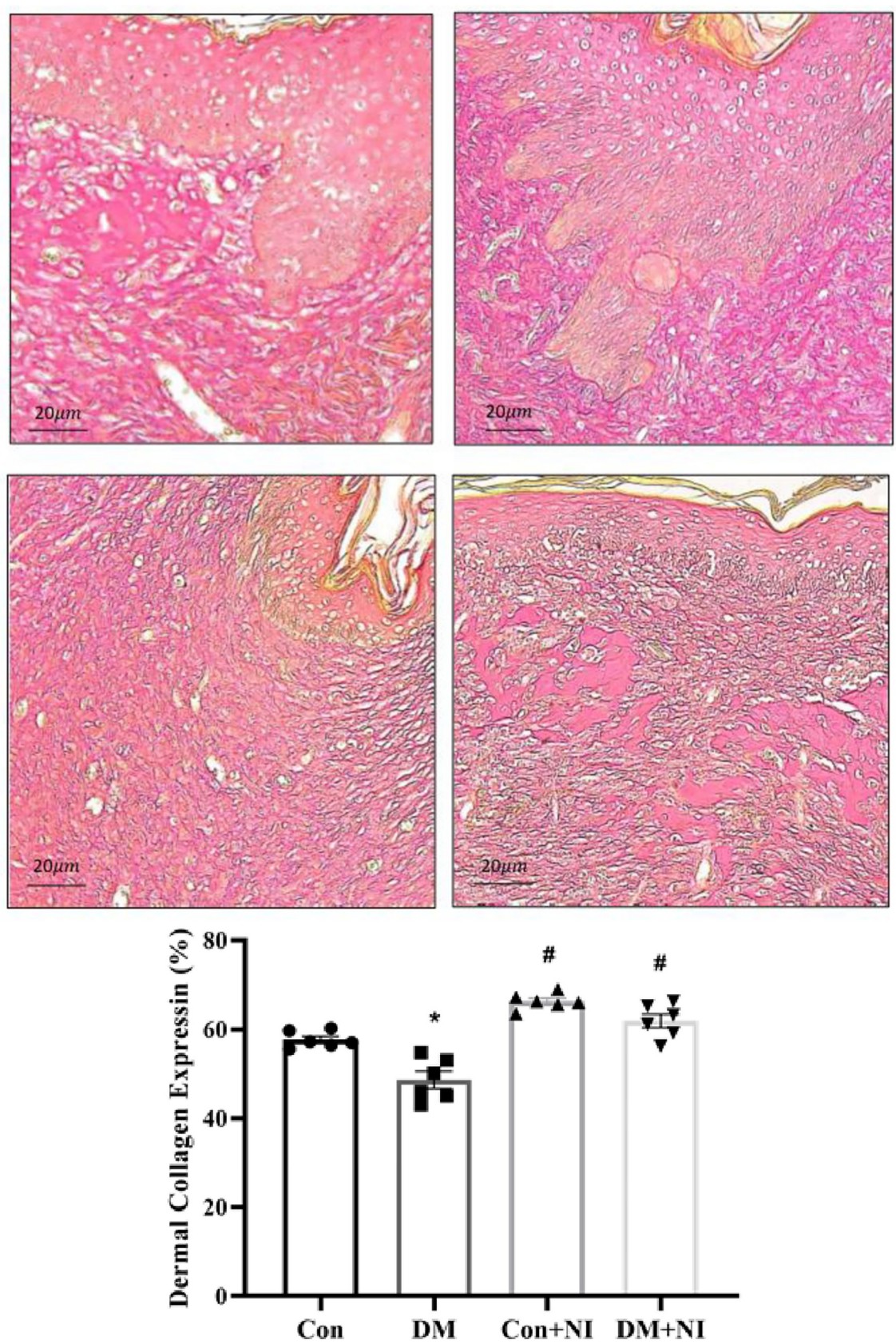

**Fig 10.** A: Representative histological photographs of wounded skin neighbor tissue stained with Sirius red staining are shown (Scale bars: 20 μm). B: Quantitative histological evaluation of wounded skin neighbor skin collagen fiber expression at day 21 after wound creation. Data are expressed as mean ± SEM (n = 6). * $P < 0.05$ vs. Con group. # $P < 0.05$ vs. DM group.

dysfunction associated with pathological features like pulmonary capillary endothelial cell damage, alveolar epithelial cell injury, extensive aqualung, tiny pulmonary atelectasis, microthrombi and microcirculation disturbance [10]. Our ALI model by LPS was consistently displayed these pathological damages as described by Zhao et al. [10]. Importantly, our present data showed that NAIs exposure in Con or LPS-induced ALI rat model did not affect the basic mean arterial blood pressure and respiratory frequency and also indicated their safety in cardiorespiratory function. To explore the NAIs effect on LPS-induced ALI, our results demonstrated that LPS increased inflammatory leukocytes infiltration into lung, and enhanced caspase 3/PARP/TUNEL-mediated apoptosis signaling in the damaged lung. However, NAIs exposure decreased the degree of inflammatory leukocytes infiltration and caspase 3/PARP/TUNEL-mediated apoptosis expression in the LPS induced ALI. Our data also evidenced that LPS depressed the possible protective signaling pathway of Beclin-1/LC3-II mediated autophagy in the lung. Interestingly, NAIs exposure significantly preserved LPS decreased Beclin-1/LC3-II mediated autophagy signaling implicating NAIs' protective effect through the action of increased autophagy signaling. Recently, effective antiviral therapeutics are urgently required to fight SARS and SARS-CoV. Our recent report found that catechins significantly reduced LPS-induced cytokine storm and oxidative stress and ALI by inhibiting PI3K/AKT/mTOR signaling to upregulate Beclin-1/Atg5-Atg12/LC3-II-mediated autophagy mechanism [16]. Liu et al. [33] demonstrated that the NAIs-related metabolic changes reflect the reduction of

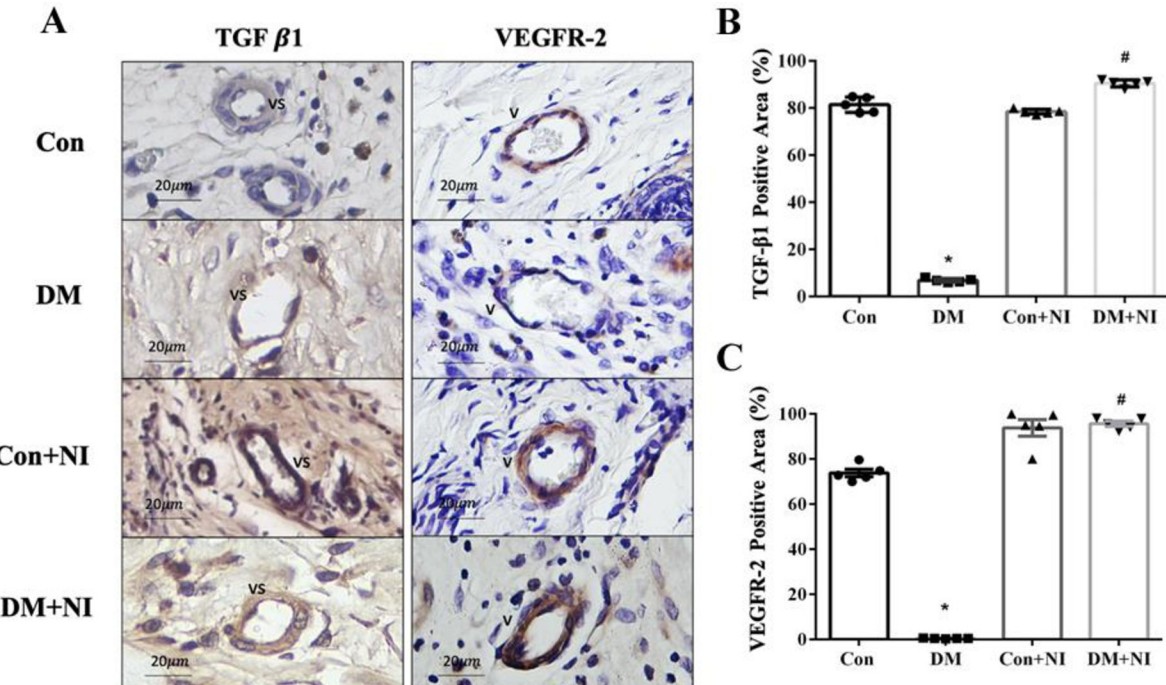

**Fig 11. Effect of NAIs exposure on rat skin TGFβ1 and VEGFR-2 expression in the wound healing of four groups of rats.** A: Representative images of TGFβ1 and VEGFR-2 immunohistochemistry on wound sections at day 21. B: Analyses of TGFβ1 stain of various treatment groups on day 21 (n = 5). C: Analyses of VEGFR-2 stain of various treatment groups on day 21 (n = 5). Data are expressed as mean ± SEM (n = 5). * $P < 0.05$ vs. Con group. # $P < 0.05$ vs. DM group.

inflammation and oxidative stress, because short-term exposure to NAIs is beneficial to heart rate variability, especially to the parasympathetic nerve activity, by successively disturbing different metabolic pathways which mainly reflect the increased anti-inflammation and the reduced inflammation. NAIs with antioxidant and anti-inflammatory activity also can attenuate LPS induced leukocyte infiltration mediated inflammation and apoptosis production possibly through the protective action of autophagy activation. Recently, in the report of Zhang et al. [34], NAIs can 100% inhibit the aerosol transmission of SARS-CoV-2 and influenza A virus. In addition, the safety of 4-week NAIs exposure for Balb/c mice showed no abnormalities in body weight, blood routine analysis, and lung pathology [34]. All these studies demonstrate that NAIs can be used as a safe and effective means of blocking respiratory virus transmission and contribute to pandemic prevention and control.

Wound healing is a complex biological process that includes blood clotting, inflammation, fibrosis, collagen deposition, and wound contraction [35]. Poorer healing in diabetics can also be observed in animal models of DM, where poor wound healing is found in mice with hereditary DM and in streptozotocin-induced diabetic rats [36]. In diabetes in the human and animal models, the research has identified a large number of changes associated with diabetes at the molecular level in delayed wound healing and to a lesser extent in chronic diabetic ulcers [37]. Oxidative stress is believed to play an important role in the development of diabetic complications [38]. Increased reactive oxygen species level could be generated in diabetes through advanced glycation end-product formation by glycation, together with glucose autoxidation and stimulation of the polyol pathway [37]. Disruption of the redox balance and alterations to the level or activity of reducing enzymes could contribute to poor healing and may even lead to damage to DNA within cells involved in the normal healing process. On the other hand, excess oxidative stress may result in tissue injury and lead to tissue in a hypoxic state. Hypoxia is an important activator of the endothelial cells in the injured vasculature [39]. During the hypoxic environment, the neutrophils, the first responders, infiltrate to the site of injury in the acute phase of inflammation [40]. Our present results from H&E stains also demonstrated that large amount of leukocytes infiltration in the DM wound skin implicating the inflammatory response in DM. The increased leukocytes infiltration also evidenced the enhanced oxidative stress in the injured tissue.

Importantly, wound monitoring should achieve rapid wound closure with functional tissue and minimal scarring. Compared with DM group with less wound healing rate, the NAIs exposure conferred significantly better effects on wound contraction. In the NAIs-treated wounds, obviously reduced the number of leukocytes infiltration and the degree of erythrocyte extravasation were observed on days 21. This indicated accelerated wound healing via the anti-inflammatory and antioxidant activities of NAIs. A recent report has indicated NAIs exerted anti-inflammatory and antioxidant effects in HaCaT cells exposed to PM implicating its application to the prevention and treatment of inflammatory skin diseases [19]. The use of antioxidant and anti-inflammatory strategies through the reduction of neutrophil adhesion and leukocyte recruitment, or inhibition of proinflammatory cytokines release could promote wound healing [25]. Although the mechanism remains uncertain how NAIs reduce inflammation and promote wound healing, our study evidenced that the NAIs exposure confers antioxidant capability, reduces RBC accumulation and leukocyte infiltration number (inflammatory index), accelerates angiogenesis from the results of increased TGF-β1 and VEGFR-2 expression in vascular beds (angiogenesis index) and collagen deposition by increased Masson's stain and Sirius red stain in the increased granular tissue, resulting in enhanced re-epithelialization. Collagen acts as a structural support for skin and helps in functioning the cell migration, maintaining the cell shape and inducing protein synthesis. One previous report stated that in the proliferative phase, fibroblasts produce collagen which replaces the fibronectin-fibrin matrix

that produce structure to the wound and in the remodeling phase, myofibroblasts deposit the collagen by cross linking in the wound which helps in wound contraction [41]. The increased collagen secretion by NAIs exposure may inhibit collagenase activity and elevate the deposition of the extracellular matrix proteins with increased fibroblast proliferation [42]. The possible mechanisms could be due to the increased collagen deposition to enhance wound contraction, and the enhanced proliferation of the various cells comprising the granular tissue by angiogenesis [25, 43]. Several growth factors like TGF-β1 and VEGF released from blood plasma or exudate in the wound and enhanced cell proliferation, migration and angiogenesis [24, 44]. TGF-β1 and VEGFR-2 have been involved in the improvement of wound repair via increases in fibroblast repopulation and angiogenesis [25]. NAIs exposure also enhanced the expression of TGF-β1 and VEGFR-2 around the vessels of skins, which appeared to mediate by VEGF regulation. Our data supported the hypothesis that NAIs exposure is able to promote TGF-β1 and VEGF dependent angiogenesis and proliferation pathways in the wound healing process. However, the exact mechanism requires further study to determine.

## 5. Conclusions

NAIs with antioxidant and anti-inflammatory effects could attenuate LPS induced ALI possibly through the enhancement of LC3-II-mediated autophagy signaling and the inhibition of caspase 3/PARP/TUNEL/apoptosis signaling in the damaged lung. NAIs also promoted wound healing possibly through the enhancement of skin collagen synthesis, vascular VEGFR-2 and TGFβ-1 mediated angiogenesis pathways.

## Supporting information

**S1 Raw images.**
(PDF)

## Acknowledgments

We thank the Ministry of Science and Technology of ROC (MOST-107-2218-E-003-001) to support this work.

**Institutional Review Board Statement:**
All surgical and experimental procedures were all approved by the Institutional Animal Care and Use Committee of National Taiwan Normal University, with Approval Number 110012 (on the date of 2021 August 5).

## Author Contributions

**Conceptualization:** Chiang-Ting Chien.

**Data curation:** Yu-Hsuan Cheng, Hung-Keng Li, Chien-An Yao, Jing-Ying Huang, Yi-Ting Sung, Shiu-Dong Chung.

**Formal analysis:** Yu-Hsuan Cheng, Hung-Keng Li, Chien-An Yao, Jing-Ying Huang, Yi-Ting Sung, Shiu-Dong Chung.

**Funding acquisition:** Shiu-Dong Chung, Chiang-Ting Chien.

**Investigation:** Yu-Hsuan Cheng, Hung-Keng Li, Chien-An Yao, Shiu-Dong Chung.

**Methodology:** Yu-Hsuan Cheng, Hung-Keng Li, Chien-An Yao, Jing-Ying Huang, Yi-Ting Sung.

**Project administration:** Shiu-Dong Chung, Chiang-Ting Chien.

**Resources:** Shiu-Dong Chung, Chiang-Ting Chien.

**Software:** Hung-Keng Li, Chien-An Yao.

**Supervision:** Chiang-Ting Chien.

**Validation:** Shiu-Dong Chung.

**Writing – original draft:** Yu-Hsuan Cheng, Hung-Keng Li, Chien-An Yao, Shiu-Dong Chung.

**Writing – review & editing:** Chiang-Ting Chien.

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
