## [Decision Letter · Decision Letter 0]

19 Jul 2022

PONE-D-22-13242Negative Air Ions through the Action of Antioxidation, Anti-inflammation, Anti-apoptosis and Angiogenesis Ameliorate Lipopolysaccharide Induced Acute Lung Injury and Promote Diabetic Wound Healing in RatPLOS ONE

Dear Dr. Chien,

Thank you for submitting your manuscript to PLOS ONE. After careful consideration, we feel that it has merit but does not fully meet PLOS ONE’s publication criteria as it currently stands. Therefore, we invite you to submit a revised version of the manuscript that addresses the points raised during the review process.

Please make sure to address the reviewers' comments especially regarding the clarity of the experiments (details, reagents, and designs), updated references, and in-depth discussion.

We look forward to receiving your revised manuscript.

Kind regards,

Y. Peter Di, Ph.D.

Academic Editor

PLOS ONE

Journal Requirements:

2. To comply with PLOS ONE submissions requirements, in your Methods section, please provide additional information on the animal research and ensure you have included details on (1) methods of sacrifice, (2) methods of anesthesia and/or analgesia, and (3) efforts to alleviate suffering.

4. PLOS ONE now requires that authors provide the original uncropped and unadjusted images underlying all blot or gel results reported in a submission’s figures or Supporting Information files. This policy and the journal’s other requirements for blot/gel reporting and figure preparation are described in detail at https://journals.plos.org/plosone/s/figures#loc-blot-and-gel-reporting-requirements and https://journals.plos.org/plosone/s/figures#loc-preparing-figures-from-image-files. When you submit your revised manuscript, please ensure that your figures adhere fully to these guidelines and provide the original underlying images for all blot or gel data reported in your submission. See the following link for instructions on providing the original image data: https://journals.plos.org/plosone/s/figures#loc-original-images-for-blots-and-gels. In your cover letter, please note whether your blot/gel image data are in Supporting Information or posted at a public data repository, provide the repository URL if relevant, and provide specific details as to which raw blot/gel images, if any, are not available. Email us at plosone@plos.org if you have any questions

5. We noticed you have some minor occurrence of overlapping text with the following previous publication(s), which needs to be addressed:

- https://onlinelibrary.wiley.com/doi/abs/10.1111/j.1464-5491.2006.01773.x

- https://tessera.spandidos-publications.com/10.3892/etm.2017.4534?text=fulltext

- http://www.ionisation.be/Effets_benefiques_de_l_ionisation_negative_de_l_air_Revue_de_70_publications_scientifiques_Mars_2011.pdf

- https://pubmed.ncbi.nlm.nih.gov/32848396/

- https://anndermatol.org/DOIx.php?id=10.5021%2Fad.2021.33.2.116

In your revision ensure you cite all your sources (including your own works), and quote or rephrase any duplicated text outside the methods section. Further consideration is dependent on these concerns being addressed.

Additional Editor Comments (if provided):

Please make sure to address the reviewers' comments especially regarding the clarity of the experiments (details, reagents, and designs), updated references, and in-depth discussion.

Reviewers' comments:

Reviewer's Responses to Questions

**Comments to the Author**

1. Is the manuscript technically sound, and do the data support the conclusions?

Reviewer #1: Yes

Reviewer #2: Partly

2. Has the statistical analysis been performed appropriately and rigorously? 

Reviewer #1: Yes

Reviewer #2: Yes

3. Have the authors made all data underlying the findings in their manuscript fully available?

Reviewer #1: Yes

Reviewer #2: Yes

4. Is the manuscript presented in an intelligible fashion and written in standard English?

Reviewer #1: Yes

Reviewer #2: Yes

5. Review Comments to the Author

Reviewer #1: In the present manuscript entitled: Negative Air Ions through the Action of Antioxidation, Anti-inflammation, Anti-apoptosis and Angiogenesis Ameliorate Lipopolysaccharide Induced Acute Lung Injury and Promote Diabetic Wound Healing in Rat, the authors focused to investigate the impact of NAI on LPS-induced acute lung injury and on the healing process of skin wounds in STZ-induced diabetic rats. Several points should be seriously taken in consideration for the following raisons:

1- The authors should alphabetically arrange the key words.

2- The authors should respect the use of abbreviations throughout the manuscript. The abbreviation must be defined at the first appear in the text and then the authors should use the abbreviations throughout the manuscript.

3- In the materials and methods section, the authors should add the information about the primary and secondary antibodies used for IHC analysis.

4- The discussion section must be improved, and the authors must add more recent references.

5- In the figure 1B &C and Figure 2B & C, the authors should add symbol for significance.

6- In the figure 5, the authors must normalize the phosphor-p65 to the total p65 not to beta-actin. Similarly, the authors must normalize the phosphor-IkB to the total IkB not to beta-actin.

7- In the figure 3A, the expression levels of cleaved caspase-3 must be normalized to the total caspase-3 not to beta-actin.

8- In figure 9 the authors used Masson’s trichrome stain to monitor collagen expression in the wounded skin tissues, but I think Sirius red stain is more suitable for this purpose.

Reviewer #2: The authors explored the benefits of negative air ions on rats in terms of acute inflammatory injury and chronic skin recovery, respectively. Although the experiments look interesting, some logical flaws and unclear expressions were presented in the manuscript. Thus, I would recommend the authors address the following comments before further consideration:

1. Figure 1B-C, the authors are trying to detect ROS level by using chemiluminescence analyzer, and the positive control group and/or negative control group of ROS level should be presented to show the accuracy of this test. For example, recognized antioxidants are recommended to use. Why did the saline group show very high levels of ROS?

2. Line 86, please make sure the detail of data from reference is correct. It is “320-350 000 ions/cm2” in the paper (Sirota et al. 2008) instead of “320-35 million ions/cm3”.

3. Line 90-99, the authors seem to want to explore COVID-19-induced ALI, so why not use SARS-CoV-2 spike protein instead of bacterial LPS to stimulate rats?

4. The authors did not mention the time of infection in the acute lung injury experiment, which is very confusing. Figure 3-5, the time frame is not shown for the changes in the protein levels following the LPS infection.

5. In Figure 5, it is very arbitrary for the authors to determine the level of inflammation in a lung by quantifying the number of cells in the pathology images. Instead of doing this, the authors should have collected alveolar lavage fluid and then quantified the level of inflammation in the lungs by the number of inflammatory cells.

6. In Figure 6, the expression of Beclin-1 was not significantly suppressed in the LPS group vs. Con group, which is contrary to the author's description (Line 294-295).

7. The pathway proteins in this study are superficial and part of the conclusions cannot be supported by the current data.

6. PLOS authors have the option to publish the peer review history of their article (what does this mean?). If published, this will include your full peer review and any attached files.

Reviewer #1: **Yes: **Gamal Badr

Reviewer #2: No

---

## [Author Response · Author response to Decision Letter 0]

8 Sep 2022

Response to the Reviewer #1:

In the present manuscript entitled: Negative Air Ions through the Action of Antioxidation, Anti-inflammation, Anti-apoptosis and Angiogenesis Ameliorate Lipopolysaccharide Induced Acute Lung Injury and Promote Diabetic Wound Healing in Rat, the authors focused to investigate the impact of NAI on LPS-induced acute lung injury and on the healing process of skin wounds in STZ-induced diabetic rats. Several points should be seriously taken in consideration for the following raisons:

1- The authors should alphabetically arrange the key words.

Ans: Thanks for your comments. We have rearranged the key words alphabetically.

2- The authors should respect the use of abbreviations throughout the manuscript. The abbreviation must be defined at the first appear in the text and then the authors should use the abbreviations throughout the manuscript.

Ans: Thanks for your comments. We have carefully used the abbreviations throughout the revised manuscript.

3- In the materials and methods section, the authors should add the information about the primary and secondary antibodies used for IHC analysis.

Ans: Thanks for your comments. We have added the information about the primary and secondary antibodies used for IHC analysis.

These messages have been included in the revised methods as following:

“We assessed the degree of apoptosis by evaluating the expression of caspase 3 and PARP and terminal deoxynucleotidyl transferase-mediated nick-end labeling (TUNEL) assay (DNA Fragmentation Kit; BioVision, Milpitas, California, USA) and autophagy by Beclin-1 (Cell Signaling Technology, Inc.) and LC3-II (Cell Signaling Technology, Inc.) staining in the rat lungs. The determined techniques have been followed as reported previously [Yang et al., 2022]. In brief, tissue sections from 10% formalin fixation and paraffin embedding were deparaffinized, rehydrated, and stained with H&E or immunohistochemically. For autophagy or apoptosis related proteins staining, the 5-µm tissue sections (Leica RM 2145, Nussloch, Germany) were incubated overnight at 4°C with mouse anti-rat Beclin-1 antibody (BD Biosciences, San Jose, CA, 1:100) or LC3-II (Cell Signaling Technology, Inc., 1:100), Caspase 3 (CPP32/Yama/Apopain, Upstate Biotechnology, Lake Placid, NY, 1:100), PARP (Cell Signaling Technology, Inc., 1:100). Subsequently, biotinylated secondary antibodies (Dako, Botany, NSW, Australia) were applied, followed by incubation with streptavidin-conjugated horseradish-peroxidase (Dako). The chromogen used in this study was Dako Liquid diaminobenzidine. Twenty high-power (�400) fields were randomly selected from each gastric section, and the value of brown deposits/total section area for Beclin-1, LC3-II, caspase-3 or PARP stain was analyzed with Adobe Photoshop 7.0.1 image software. TUNEL was performed according to a previously described method [Yang et al., 2008]. Briefly, 5-µm-thick sections of gastric tissues were prepared, deparaffinized, and stained using the TUNEL assay kit.” (Line)

“In brief, tissue sections from 10% formalin fixation and paraffin embedding were deparaffinized, rehydrated, and stained with VEGFR-2 or TGF-β1. For VEGFR-2 or TGF-β1 staining, the 5-µm tissue sections were incubated overnight at 4°C with mouse anti-rat VEGFR-2 antibody (VEGF #9698 Cell Signaling Technology, Danvers, MA, USA, 1:100) or TGF-β1 (sc-146 Santa Cruz, Dallas, Texas, USA, 1:100). Next, biotinylated secondary antibodies (Dako, Botany, NSW, Australia) were applied, followed by incubation with streptavidin-conjugated horseradish-peroxidase (Dako). The chromogen used in this study was Dako Liquid diaminobenzidine. Twenty high-power (�400) fields were randomly selected from each section, and the value of brown deposits/total section area for VEGFR-2 or TGF-β1 stain was analyzed with Adobe Photoshop 7.0.1 image software as described above.” (Lines 197-210)

4- The discussion section must be improved, and the authors must add more recent references.

Ans: Thanks for your comments. We have rewritten the discussion section with red color and included more recent references.

5- In the figure 1B &C and Figure 2B & C, the authors should add symbol for significance.

Ans: Thanks for your comment. The Figure 1B is an original graph, therefore, no statistic symbol could be added. We have added the symbol for significance in the Figure 1C and Figures 2B & 2C. 

6- In the figure 5, the authors must normalize the phosphor-p65 to the total p65 not to beta-actin. Similarly, the authors must normalize the phosphor-IkB to the total IkB not to beta-actin.

Ans: Thanks for your comment. We have no data involving phosphor-p65 and phosphor-IkB in this experiment. 

7- In the figure 3A, the expression levels of cleaved caspase-3 must be normalized to the total caspase-3 not to beta-actin.

Ans: Thanks for your comment. We have used the total caspase-3 to replace �-actin in the new Figure 3A. 

8- In figure 9 the authors used Masson’s trichrome stain to monitor collagen expression in the wounded skin tissues, but I think Sirius red stain is more suitable for this purpose.

Ans: Thanks for your comments. We have included the Sirius red stain in new Figure 10.

The information of Sirius red stain has been indicated in the Methods, Results and Discussion sections of the revised manuscript.

 

Response to the Reviewer #2:

Reviewer #2: The authors explored the benefits of negative air ions on rats in terms of acute inflammatory injury and chronic skin recovery, respectively. Although the experiments look interesting, some logical flaws and unclear expressions were presented in the manuscript. Thus, I would recommend the authors address the following comments before further consideration:

1. Figure 1B-C, the authors are trying to detect ROS level by using chemiluminescence analyzer, and the positive control group and/or negative control group of ROS level should be presented to show the accuracy of this test. For example, recognized antioxidants are recommended to use. Why did the saline group show very high levels of ROS?

Ans: Thanks for your comments. We used saline containing H2O2 as positive control and saline with vitamin C containing H2O2 as negative control in the revised Figure 1.

In our methods, 200 µL of saline with or without NAIs treatment was mixed with 0.5 mL of 0.1 mmol/L luminol (5-amino-2,3-dihydro-1,4-phthalazinedione, Sigma, Chemical Co., St. Louis, MO, USA) and 0.1 mL of H2O2 (0.03%) and was analyzed with a chemiluminescence analyzing system (CLA-ID3, Tohoku Electronic Inc. Co., Sendai, Japan). The chemiluminescence signals emitted from the mixture of saline, H2O2 and luminol, which represented the hydrogen peroxide content in the mixture, were recorded. The increased chemiluminescent signals from the sample–luminol–H2O2 mixture were determined for 300 s. Therefore the saline group showed a very high level of ROS for the containing H2O2 (0.03%).

2. Line 86, please make sure the detail of data from reference is correct. It is “320-350 000 ions/cm2” in the paper (Sirota et al. 2008) instead of “320-35 million ions/cm3”.

Ans: Thanks for your comment. We have corrected the data “320-350 000 ions/cm2” in the revised text.

3. Line 90-99, the authors seem to want to explore COVID-19-induced ALI, so why not use SARS-CoV-2 spike protein instead of bacterial LPS to stimulate rats?

Ans: Thanks for your comments. We have included new reference to state that “SARS-CoV and SARS-CoV-2 could induce acute lung injury (ALI) and inflammation, which is mimicked by lipopolysaccharide (LPS), a bacterial endotoxin present on the outer membrane of Gram-negative bacteria [Pooladanda et al., 2021], implicating SARS-CoV-2 and LPS through similar pathways of binding toll-like receptor-4 (TLR4) to activate angiotensin converting enzyme 2 (ACE2), to induce ALI and hyperinflammation [Aboudounya et al., 2021; Yang et al., 2021].” in the Introduction section. 

4. The authors did not mention the time of infection in the acute lung injury experiment, which is very confusing. Figure 3-5, the time frame is not shown for the changes in the protein levels following the LPS infection.

Ans: Thanks for your comment. The methods for the time of infection were rewritten as following. “During the stabilization period of 30 min after anesthesia, 200 μg/kg of LPS stimulation was immediately infused into trachea at 200 μl for 210 min. Physiological functions were continuously recorded for 30 min of baseline period and 210 min of experimental period, and arterial blood gases and respiratory mechanics were recorded at the end of the experiment. The NAIs treatment to the rats by 10-cm distance of NAIs exposure were performed for 210 min. Control rats were also placed indoors, but are only exposed to the air in the environmental chamber. We determined the respiratory rate, flow rate and breathing depth in these animals in total 240 min.” The protocol was also indicated in the new Figure 2A. The time frame of Figures 3-5 has also indicated in each figure legend.

5. In Figure 5, it is very arbitrary for the authors to determine the level of inflammation in a lung by quantifying the number of cells in the pathology images. Instead of doing this, the authors should have collected alveolar lavage fluid and then quantified the level of inflammation in the lungs by the number of inflammatory cells.

Ans: Thanks for your comment. We have done the larvage neutrophils and reactive oxygen species amount according to your suggestion in the new Figures 5E& 5F.

6. In Figure 6, the expression of Beclin-1 was not significantly suppressed in the LPS group vs. Con group, which is contrary to the author's description (Line 294-295).

Ans: Thanks for your comments. We have reworked a new western blot of Beclin-1 in the revised text as shown in new Figure 6. This data now is consistent to the description on Line 294-295.

7. The pathway proteins in this study are superficial and part of the conclusions cannot be supported by the current data.

Ans: Thanks for your comments. We have modified the conclusions for consistency and clarification according to the current data.

---

## [Decision Letter · Decision Letter 1]

22 Sep 2022

Negative Air Ions through the Action of Antioxidation, Anti-inflammation, Anti-apoptosis and Angiogenesis Ameliorate Lipopolysaccharide Induced Acute Lung Injury and Promote Diabetic Wound Healing in Rat

PONE-D-22-13242R1

Dear Dr. Chien,

We’re pleased to inform you that your manuscript has been judged scientifically suitable for publication and will be formally accepted for publication once it meets all outstanding technical requirements.

Kind regards,

Y. Peter Di, Ph.D.

Academic Editor

PLOS ONE

Additional Editor Comments (optional):

Reviewers' comments:

Reviewer's Responses to Questions

**Comments to the Author**

1. If the authors have adequately addressed your comments raised in a previous round of review and you feel that this manuscript is now acceptable for publication, you may indicate that here to bypass the “Comments to the Author” section, enter your conflict of interest statement in the “Confidential to Editor” section, and submit your "Accept" recommendation.

Reviewer #1: All comments have been addressed

Reviewer #2: All comments have been addressed

2. Is the manuscript technically sound, and do the data support the conclusions?

Reviewer #1: Yes

Reviewer #2: Yes

3. Has the statistical analysis been performed appropriately and rigorously? 

Reviewer #1: Yes

Reviewer #2: Yes

4. Have the authors made all data underlying the findings in their manuscript fully available?

Reviewer #1: Yes

Reviewer #2: Yes

5. Is the manuscript presented in an intelligible fashion and written in standard English?

Reviewer #1: Yes

Reviewer #2: Yes

6. Review Comments to the Author

Reviewer #1: The authors properly answered all the raised comments and the revised manuscript is now suitable for publication.

Reviewer #2: (No Response)

7. PLOS authors have the option to publish the peer review history of their article (what does this mean?). If published, this will include your full peer review and any attached files.

Reviewer #1: **Yes: **Gamal Badr

Reviewer #2: **Yes: **Zhonghui Zhu

---

## [Editor Report · Acceptance letter]

13 Oct 2022

PONE-D-22-13242R1 

Negative Air Ions through the Action of Antioxidation, Anti-inflammation, Anti-apoptosis and Angiogenesis Ameliorate Lipopolysaccharide Induced Acute Lung Injury and Promote Diabetic Wound Healing in Rat 

Dear Dr. Chien:

I'm pleased to inform you that your manuscript has been deemed suitable for publication in PLOS ONE. Congratulations! Your manuscript is now with our production department. 

Kind regards, 

on behalf of

Dr. Y. Peter Di 

Academic Editor

PLOS ONE